# Heat Treatment in Two Tomato Cultivars: A Study of the Effect on Physiological and Growth Recovery

**Sherzod Nigmatullaevich Rajametov [1], Eun Young Yang [1,\*], Hyo Bong Jeong [1,†], Myeong Cheoul Cho [1,†], Soo Young Chae [1] and Niroj Paudel [1,2]**

[1] Vegetable Research Division, Department of Horticultural Crop Research, National Institute of Horticultural & Herbal Science, Rural Development Administration, Wanju 55365, Korea; sherzod_2004@list.ru (S.N.R.); bong9846@korea.kr (H.B.J.); chomc@korea.kr (M.C.C.); cotez@korea.kr (S.Y.C.); nirojjirauna@gmail.com (N.P.)

[2] Department of Applied Plant Science, Kangwon National University, Chuncheon 24341, Korea

\* Correspondence: yangyang2@korea.kr; Tel.: +82-63-238-6613; Fax: +82-63-238-6605

† Equal contribution.

**Abstract:** High temperature (HT) significantly affects crop physiological traits and reduces productivity in plants. To increase yields as well as survival of crops under HT, developing heat-tolerant plants is one of the main targets in crop breeding programs. The present study attempted to investigate the linkage of the heat tolerance between the seedling and reproductive growth stages of tomato cultivars 'Dafnis' and 'Minichal.' This research was undertaken to evaluate heat tolerance under two experimental designs such as screening at seedling stage and screening from reproductive traits in greenhouses. Survival rate and physiological responses in seedlings of tomatoes with 4-5 true leaves were estimated under HT (40 °C, RH 70%, day/night, respectively) and under two control and HT greenhouse conditions (day time 28 °C and 40 °C, respectively). Heat stress significantly affected physiological–chemical (photosynthesis, electrolyte conductivity, proline) and vegetative parameters (plant height, shoot fresh weight, root fresh weight) in all tomato seedlings. The findings revealed that regardless of tomato cultivars the photosynthesis, chlorophyll, total proline and electrical conductivity parameters were varied in seedlings during the heat stress period. The heat tolerance rate of tomatoes in the seedling stage might not always be associated with reproductive parameters. HT reduced fruit parameters such as fruit weight (31.9%), fruit length (14.1%), fruit diameter (19.1%), and fruit hardness (9.1%) compared to NT under HT in heat-susceptible tomato cultivar 'Dafnis', while in heat-tolerant cultivar 'Minichal' fruit length (7.1%) and fruit diameter (12.1%) was decreased by the effects of HT, but on the contrary fruit weight (3.6%) and fruit hardness (8.3%) were increased. In conclusion, screening and selection for tomatoes should be evaluated at the vegetative and reproductive stages with consideration of reproductive parameters.

**Keywords:** tomato; high temperature; damage; seedling; root; weight; flower; fruit; photosynthesis; proline; electrolyte conductivity

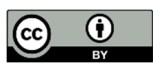

## 1. Introduction

Temperature stress has become and will continue to be a great concern in agriculture cultivation due to climate change. Crops, including tomato (*Solanum lycopersicum* L.) cultivars, have a narrow range of optimal growing temperatures ranging from 25 to 30 °C during the daytime and 20 °C at night [1,2] and are affected by both high [2–4] and low-temperature stress [5–7]. Due to intensive breeding of a few desired traits during domestication, the genetic diversity of commercial tomato cultivars has declined, whereas wild species have still maintained a larger number of valuable traits [8]. High temperature and

lack of tolerant cultivars have inhibited to increase of the cultivation area of tomato cultivars, due to adverse effect of the temperature on morphological-physiological, reproductive and  yieldproperties [3,9–12].

The area of tomato cultivation has been increasing around the world annually and reaching more than 5 million ha, producing 180,766,329 metric tons in 2019, whereas in South Korea its cultivation area and production were 6460 ha and 420,573 metric tons, respectively (http://www.fao.org/faostat/, accessed on 17 May 2021). The importance of tomato plants in agricultural crops has certainly emerged. However, while reproductive tolerance during heat stress is an important value for the evaluation of tomato cultivars yield [13–16], tomatoes, in vegetative and reproductive stages, are sensitive to high temperature and have varying sensitivity to stress [17,18].

Many research works were conducted to evaluate heat tolerance and understand mechanism and physiological responses to high temperature in tomatoes to identify a tolerant specimen [2,12,18]. The yield traits of tomatoes depending on the growth conditions are noticeably varied in one genotype [16]. Recently, multiple methods to screen for heat-tolerant tomato plants were validated at different growth stages under heat stress conditions [2–4,15]. The responses of different traits to the high temperature varied, and the results of correlation analysis showed the relationships between various traits (pollen viability, fruit set, flower number per inflorescence) within the control and heat-treated plants, but not between the two [14]. Tomato plants are sensitive to high temperature and displayed diverse responses to stress during the vegetative and reproductive stages [17,18]. Reproductive traits including the number of flowers (NFL), fruits (NFR), and fruit set (FS) during heat stress are important values for evaluating the good yields of tomato cultivars [13–16].

However, in tomatoes, the underlying mechanisms to abiotic and biotic stresses are not well understood even today [14,19]. Therefore studies on elucidating the mechanisms of high temperature tolerance in the seedling stage of tomatoes and investigation of linkage in heat-tolerant in main traits such as fruit set, yield, and fruit size are important [14–16].

The purpose of this research is to analyze the survival rates and physiological responses in the seedlings as well as adult plants of two tomato cultivars with contrasting heat tolerance levels.

## 2. Materials and Methods

Experiment I 'Screening of heat tolerance of tomato cultivars at seedling stage'.

### 2.1. Plant Materials and Heat Treatment Conditions

The seeds of commercial tomato cultivars 'Dafnis' (D) and 'Minichal' (M), which are widely cultivated in South Korea, were sown in plastic trays (52 × 26 cm in size, 6 × 6 cm cells with pot volume 5 L) containing a 1:1 ratio of sand and commercial bed soil (Bio Sangto, Seoul, Korea) consisting of coco peat (47.2%), peat moss (35%), zeolite (7%), vermiculite (10.0%), dolomite (0.6%), humectant (0.006%), and fertilizers (0.194%). The trays were watered with 1 L daily, and placed in a glasshouse (28/18 °C in day/night with relative humidity within 65–70%) in the National Institute of Horticultural and Herbal Science, South Korea. Tomato seedlings with 4–5 true leaves (4LS) on 30 days after sowing were transferred to a growth chamber for heat treatment. The seedlings were maintained under severe HT conditions (40 °C day/night, 16/8-h light/dark cycle) and light intensity of 800 μmol m$^{-2}$ s$^{-1}$ within 70% relative humidity. For each cultivar, 4 technical replications (a total of 32 seedlings) were heat-treated in the growth chamber for 7 days and watered twice a day with a total of two liters to avoid drought stress. After HT treatment, the seedlings were transferred to normal conditions (28/18 °C, day/night) and maintained for 3 days.

### 2.2. Measurement of Heat Tolerance among Tomato Seedlings

Leaf heat damage levels (LHD) of heat-treated tomato plants after 7 days of HT treatment were identified according to the visual injuries. Leaf damage was investigated by measuring the percentage of leaf area that was dried or light yellow-white colored. LHD was classified into four levels: LHD 0% (not heat-treated), LHD 25% (leaf damages from 11 to 25%), LHD 50% (leaf damages from 25 to 50%), and LHD 75% (leaf damages from 50 to 75%). After 7 days of HT, the seedlings were transferred to the glasshouse condition as described above and maintained for 3 days to recover.

### 2.3. Measurement of Chlorophyll Contents and Photosynthetic Rate in Seedlings under Heat Treatment

Total chlorophyll index (CHL) was estimated from three independent biological replicates using SPAD meter (Konica Minolta, Japan) in tomatoes from 3rd–4th leaves on day 0 (initial rate, no treated- NT), 1, 3, 5, and day 7 of HT, respectively.

The photosynthetic rate ($\mu$mol $CO_2$ $m^{-2}s^{-1}$), stomatal conductance (mol $H_2O$ $m^{-2}s^{-1}$) intercellular $CO_2$ concentration ($\mu$mol $CO_2$ $mol^{-1}$), and transpiration rate (mmol $H_2O$ $m^{-2}s^{-1}$) were measured from 3rd–4th leaves of 0, 1, and 3 days after HT between 10:00–12:00 a.m. Data were recorded in three plants per cultivar using a portable photosynthesis measurement system (LI-6400, LI-COR Bioscience, Lincoln, NE, USA). Light response curves (PAR) were set to 800 $\mu$mol $m^{-2}s^{-1}$, the temperature of leaf chamber was set to 25 °C, and the intercellular $CO_2$ concentration was maintained at 400 $\mu$mol ($CO_2$) $mol^{-1}$. The photosynthetic rate was automatically measured at each irradiation level after 3–4 min light exposure [4,20].

### 2.4. Determination of Electrolyte Leakage Potential in Seedlings Leaves under HT

The leakage of electrolyte from tomato leaves was measured according to Camejo et al. [1] with minor modifications. Leaves from 3rd–4th nodes from seedlings (used in three technical replications) were perforated into discs with a radius of 5.5 mm. Each disc was placed in a 15-mL tube containing 10 mL of deionized water and then incubated on a shaker at 25 °C for 30 min. At this time, the conductivity (EC1) of water was measured using a STARA-HB conductivity meter (Thermo Orion, Waltham, MA, USA). The tube was heated in a boiling water bath for 30 min and cooled at room temperature for 20 min, and then the conductivity (EC2) was measured. Final EC content was expressed as the percentage of EC1/EC2.

### 2.5. Extraction of Free Total Proline Content in Seedlings Leaves under HT

Free total proline content (PRL) in tomato leaves was measured using colorimetric assay [21]. Leaf samples were prepared as mentioned above in the determination of EC. All leaves were lyophilized (−72 °C) in a freezer dryer (IlShin BioBase, Seoul, Korea) for 3 days. Each leaf sample, weighing 100 mg (dry weight), was homogenized with 2 mL of 3% (*w/v*) aqueous sulfosalicylic acid solution. The homogenate was centrifuged at 14,000 rpm for 7 min. Then 1 mL of supernatant was transferred to 5 mL microtubes, 1 mL of glacial acetic acid, and 1 mL of acid ninhydrin. The ninhydrin reaction was prepared by adding ninhydrin (2.5 g/100mL) to a solution containing glacial acetic acid, distilled water, 85% of 6 M ortho-phosphoric as a ratio of 6:3:1 receptively. Immediately the reaction mixtures were placed in a boiling water bath (95 °C) for 1 h and the reaction was stopped at 4 °C for 20 min. The reading were taken at a wavelength of 546 nm by spectrophotometer (EON, BioTek Instruments, Winooski, VT, USA).

### 2.6. Proline Content and Seedlings Growth with Different Leaf Damage Levels at Recovery

To estimate the effect of the different LHD levels on the vegetative parameters of tomatoes, the seedlings maintained during the recovery were transplanted to plastic pots with the same substrates as described above and all plants were again maintained in a

glasshouse condition (30–32/22–24 °C in day/night) for 28 days. All tomato plants were watered once a day and fertilized weekly with 1 L of water containing 1 mL of N-6, P-10, and K-5 (HYPONeXm, Osaka, Japan). Proline content of seedlings with LHD 0, 25, 50, and 75% were measured at 8 days after HT with the same methods described above.

The plant (shoot) height (PH) and biomass such as shoot fresh weight (SFW) and root fresh weight (RFW) were measured from three independent biological replicates using a ruler and electron Micro Weighing Scale MW-II (CAS), respectively.

Experiment II 'Screening of heat tolerance of tomato cultivars at reproductive stage'.

### 2.7. Plant Materials and Heat Treatment Conditions

The same set of tomato cultivars as mentioned in experiment I with 6–7 LS were transplanted at a spacing of 40 cm by 40 cm (6 biological replications per accession) into two polyethylene greenhouses, where temperature set-point for ventilation in the first week was maintained within 25 °C in both greenhouses to ensure seedlings to adapt new environment. Furthermore, day temperature set-points for ventilation were changed to 28 °C and used as a normal treatment (NT) and 40 °C for screening of high temperature (HT)-tolerant tomatoes according to reproductive traits, respectively. The soil in two greenhouses was prepared according to the recommendations of the Korea Soil Information System [22] equally with pre-plant broadcast manure at a dose of 1 kg m$^{-2}$ and basal fertilizer containing 16 g m$^{-2}$ N, 8 g m$^{-2}$ K$_2$O, 16 g m$^{-2}$ P$_2$O$_5$ and regularly watered to avoid drought and fertilized weekly (Mulpure, Daeyu Co. Ltd., Gyeongsan, Korea).

### 2.8. Data Collection on Reproductive Parameters at Growth Period

The number of flowers (NFL) and fruits (NFR), fruit set (FS) per truss, and fruit yield (FY) per plant were determined from the second to fourth trusses in 6 plants in both NT and HT greenhouses. FS (%) was calculated as follows:

$$\text{Fruit set (\%)}=\frac{\text{The number of fruits}}{\text{The number of flowers}}\times100$$

Fruit yield (FY) was determined by the sum of the fresh weight of fruits (FW) in kg harvested from the second to fourth truss of six plants. Randomly, ten tomatoes of each cultivar were collected for fruit weight, fruit length (FL), fruit diameter (FD), and fruit hardness (FH) using a digital electron Micro Weighing Scale MW-II (CAS), caliper, and Fruit Hardness Tester Cat. No. 9200, Model 1 kg, Ø0.8 mm (Tokyo, Japan), respectively

### 2.9. Statistical Analysis

The experimental design of this study was completely randomized. Statistical analysis was performed using the SAS Enterprise Guide 7.1 (SAS Institute Inc., Cary, NC, USA) to identify the significant difference in the parameters among vegetative and reproductive parameters, and mean values were compared with a significance level of 5% using Duncan's multiple range test or the Student's *t*-test at the $p \leq 0.05$, $p \leq 0.01$, and $p \leq 0.001$ levels, respectively.

## 3. Results

Experiment I.

### 3.1. Screening for Heat Tolerance in Tomato Seedlings

Heat damage symptoms were observed on day 2 and were not significantly different among tomato seedlings in M and 'D.' Survival and leaf heat damage (LHD) rates were investigated from 3 days of exposure to heat stress and the data showed significant differences among tomato seedlings between M and D at 3, 5, and 7 days of heat treatment (Figure 1).

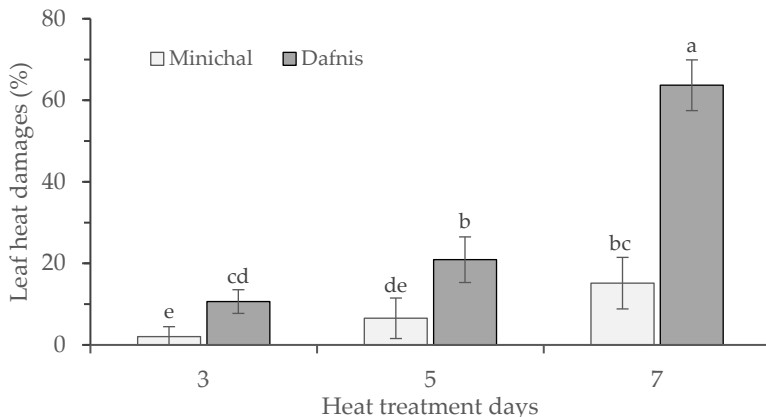

**Figure 1.** Changes in leaf heat damages among tomato cultivar seedlings Minichal and Dafnis. Vertical bars represent ± SD (n = 4). Means with different letters indicate significant differences at *p* ≤ 0.05.

Moreover, on day 7, differences in heat tolerance were significantly observed among tomato cultivars, wherein the seedlings of D were identified with high LHD (over 60%) and screened as heat-susceptible while M remained stable in its heat tolerance, with more green leaves (Figure 2).

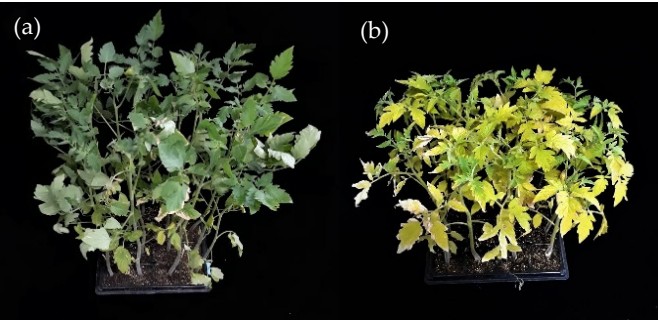

**Figure 2.** Differences in heat tolerance after 7 days of stress regime among seedlings of tomatoes Minichal (**a**) and Dafnis (**b**).

*3.2. The Difference in Physiological Responses to High Temperature between Heat-Susceptible and Tolerant Seedlings*

The result demonstrated that the chlorophyll degradation was much more prominent in the tomato seedling of D than that in M from 3 days of HT, while the chlorophyll contents in M were not significantly different from 0 to 7 days of HT (Figure 3) and the CHL of M was approximately 2 times higher than that of D.

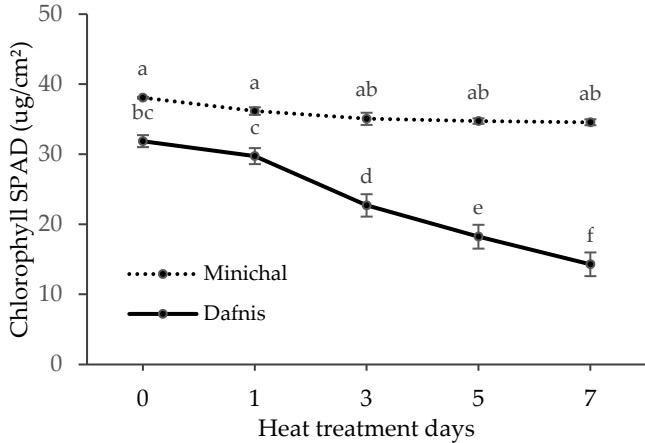

**Figure 3.** Response of chlorophyll content in tomato leaves (4LS) on heat treatment periods. Vertical bars represent ± SE (n = 3). Means with different letters indicate significant differences at $p \leq$ 0.05.

Photosynthetic parameters like $P_N$, $Gs$, $Ci$, and $Tr$ significantly varied among the tomatoes by the heat treatment days (Figure 4a–d). A steady and significant decrease in $P_N$, $Gs$, $Ci$, and $Tr$ were observed in both cultivars on day 3 of HT, where M showed the highest values.

Although the rate of $Ci$ and $Tr$ was slightly higher in D than M before HT (Figure 4c,d), the rates were steadily declining in D during HT. In addition to this, the rate of $P_N$ and $Gs$ was dramatically reduced in D, more than M (Figure 4a). Overall, the high rate of $P_N$, $Gs$, $Ci$, and $Tr$ persisted more in heat-tolerant M than heat-susceptible D on day 3 of HT (Figure 4a–d). The thermo-stability of the cell membrane was calculated by EC and the values varied among tomato seedlings on days of HT in M and D, but it was obviously higher in susceptible D than that of M during the period of HT (Figure 5a). Furthermore, the proline was well-known for the indicator of abiotic stresses such as heat stress, cold stress, and drought stress. In order to determine whether heat stress influences the accumulation of proline (PRL) content in M and D, the amount of PRL was measured during the period of HT. The PRL content of heat-susceptible D was significantly higher than that in heat-tolerant M for all the days of HT (Figure 5b).

In order to understand whether the heat stress regime was involved in vegetative parameters in M and D, plant height (PH), shoot fresh weight (SFW), and root fresh weight (RFW) were investigated. The PH, SFW, and RFW were significantly decreased in the tomato seedlings of M in HT than those in normal treatment (NT) condition (Figure 6a–c), whereas no distinct difference in PH and RFW was observed in the heat-susceptible tomato D (Figure 6a,c).

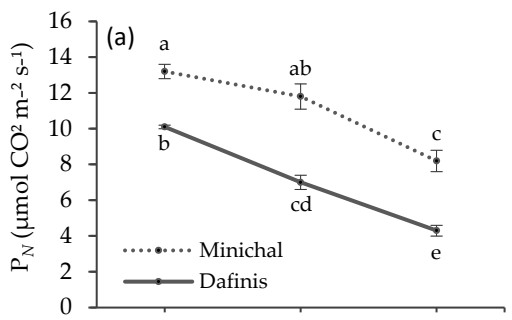

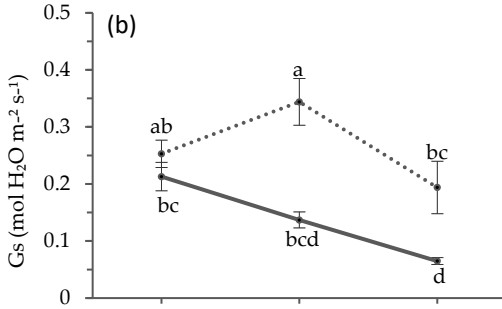

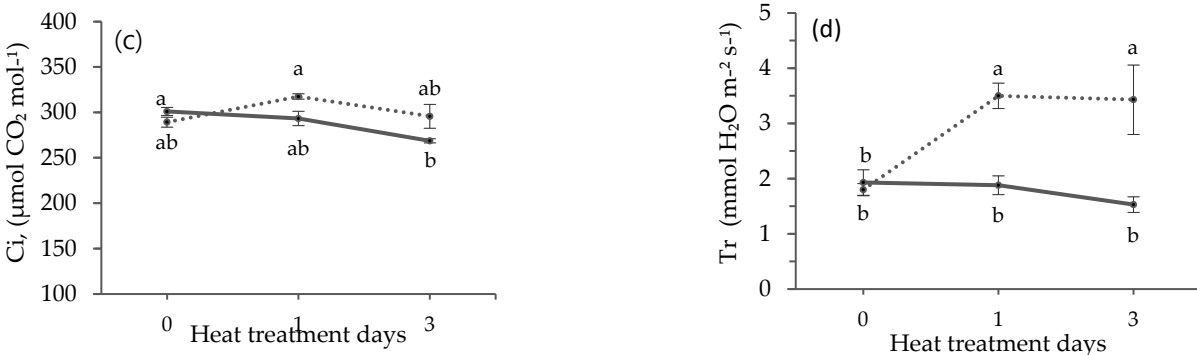

**Figure 4.** Effect of heat treatment (40 °C) on photosynthetic rate (**a**), stomatal conductivity (**b**), intercellular $CO_2$ concentration (**c**) and transpiration rate (**d**). The values are represented as means ± SE (n = 3). Different letters above bars indicate significant differences at $p \le 0.05$.

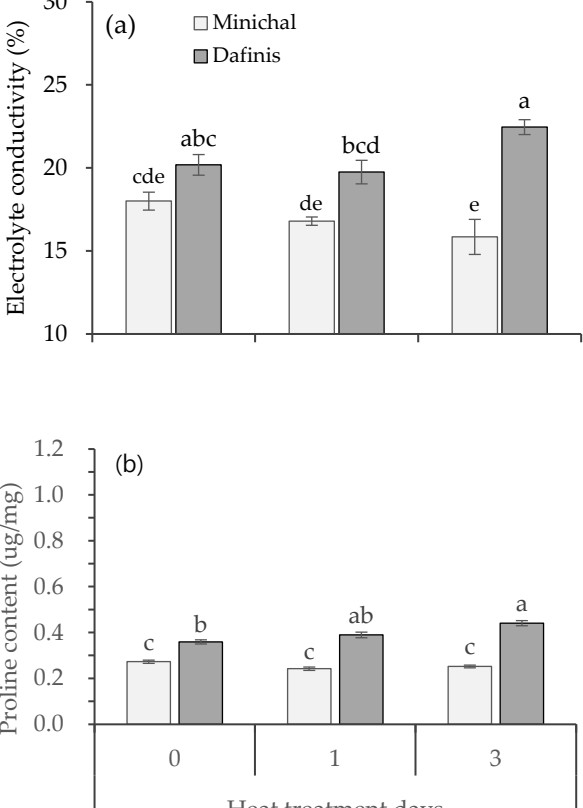

**Figure 5.** Electrolyte conductivity (**a**) and proline content (**b**) as affected by different heat treatment days in tomato seedlings of Minichal and Dafnis grown at 40 °C. Vertical bars represent ± SE (n = 3). Means with different letters indicate significant differences at $p \le 0.05$.

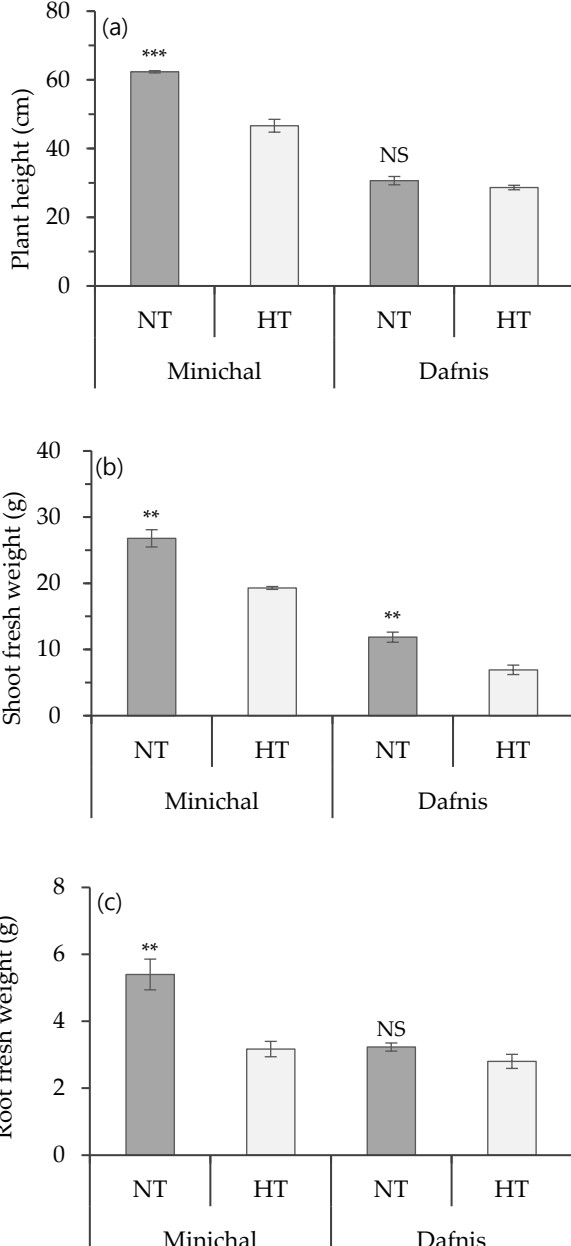

**Figure 6.** Effect of heat treatment on vegetative parameters of plant height (**a**), shoot fresh weight (**b**), and root fresh weight (**c**) in tomato seedlings of Minichal and Dafnis. NT:normal treatment, HT: heat treatment. Values are means ± SE (n = 3). NS, ** and *** indicate not significant and significant at the $p \leq 0.01$ and $p \leq 0.001$ levels in *t*-test, respectively.

*3.3. Effects of Leaf Heat Damage Levels on the Growth and Proline Content of Heat-Susceptible and Tolerant Cultivars at Recovery*

Evaluation of the growth activity in the vegetative parameters like PH, SFW, and RFW in tomato seedlings having different LHD rates showed that heat-tolerant and susceptible cultivars were significantly affected by LHD levels after recovering for 28 days (Figure 7). Plant height in both tomatoes was not significantly different in plants LHD-0, 25%, and 50% although there was an innate difference between the two cultivars, but LHD-75% significantly ($p \leq 0.05$) affected the growth rate in two cultivars (Figure 7a). Shoot fresh weight among tomato cultivars varied significantly compared to the growth rate. In a heat-susceptible D cultivar, LHD below 25% significantly affected the SFW than

compared to M plants. This was not significantly affected by LHD-25% and 50%, but it was by LHD-75% (Figure 7b). In addition, it was identified that plants with LHD significantly affected the root fresh weight among heat-tolerant and susceptible cultivars (Figure 7c).

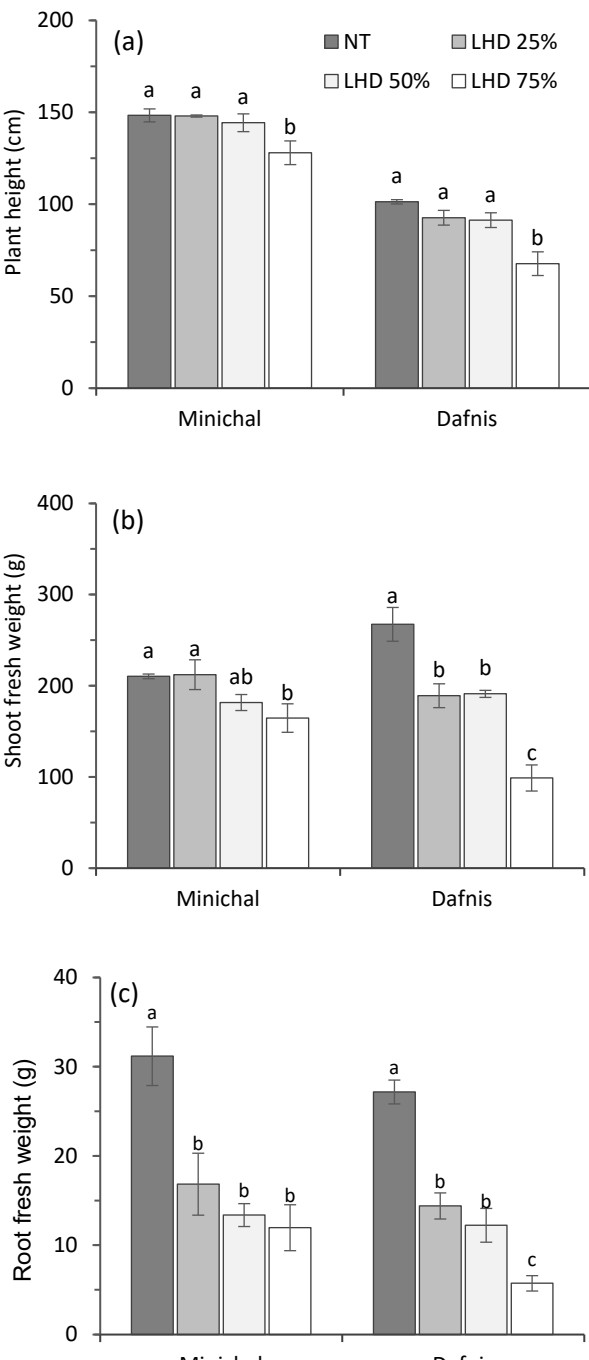

**Figure 7.** Effect of leaf heat damage levels on plant height (**a**), shoot fresh weight (**b**) and root fresh weight (**c**) in tomato cultivars. Vertical bars represent ± SE (n = 3). Values with different letters indicate significant differences at $p \leq 0.05$.

In comparison with the estimation of the accumulation of the proline content in seedling heat treatment periods, the PRL increased drastically in both cultivars as LHD levels increased from 0 to 75% in 8 days after transplanting, at recovery (Figure 8). However,

there was a significantly higher accumulation of PRL in heat-tolerant tomato M than in susceptible cultivar D, which showed contrasting values during the heat treatment of the seedlings (Figure 6b).

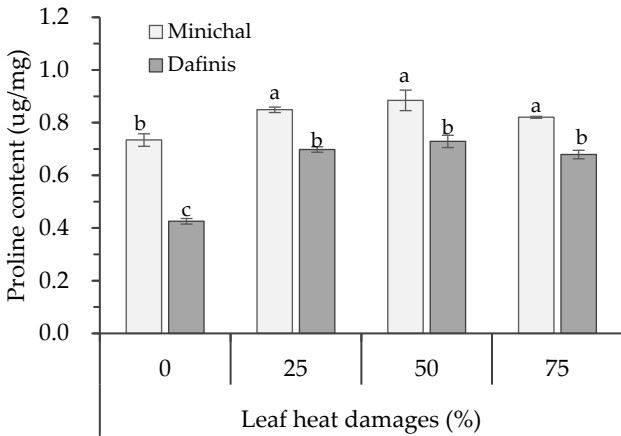

**Figure 8.** Total proline content as affected by different leaf heat damages in tomato cultivars grown for 8 days at normal condition. Vertical bars represent ± SE (n = 3). Means with different letters indicate significant differences at $p \leq 0.05$.

Experiment II

### 3.4. Effect of Heat Treatment on the Development of Flowers, Fruit Parameters, and Yield of Heat-Susceptible and Tolerant Cultivars

Determination of the effect of HT on tomato reproductive parameters development showed no significant differences in NFL between HT and NT conditions (Figure 9a). In contrast to NFL, a significant reduction in NFR was prominent in both cultivar plants under HT, showing more than 62.6% and 61.5% reduction compared to NT, respectively (Figure 9b).

The same pattern with significant differences was observed in assessment of FS, which significantly decreased in both tomato cultivars M and D under HT than in plants in NT by 58.4% and 64.1%, respectively (Figure 9c). FY was also generally reduced significantly in the two cultivars M and D at HT conditions, more than 77.3% and 60.0% reduction, respectively (Figure 9d).

High temperature significantly reduced the fruit weight, fruit length and diameter, and fruit hardness in heat-susceptible tomato cultivar D, although the same values were determined in heat-tolerant M, but excepting the fruit weight and fruit hardness (Figure 10). Significant differences ($p \leq 0.001$) were observed among FW, FL, and FD (Figure 10a–c) and in FH ($p \leq 0.01$) in the responses of tomato D to HT (Figure 10d), while, in general, both FL ($p \leq 0.05$) and FD ($p \leq 0.001$) of tomato M was significantly reduced under HT compared to NT.

Overall, in the heat-susceptible tomato cultivar D, fruit parameters FW, FL, FD, and FH decreased by 31.9%, 14.1%, 19.0%, and 9.1% under HT, respectively compared to NT, while in heat-tolerant M affected by HT, it decreased only FL and FD by 7.1% and 12.1%, respectively, but increased the FW and FH by 3.6% and 8.3%, respectively.

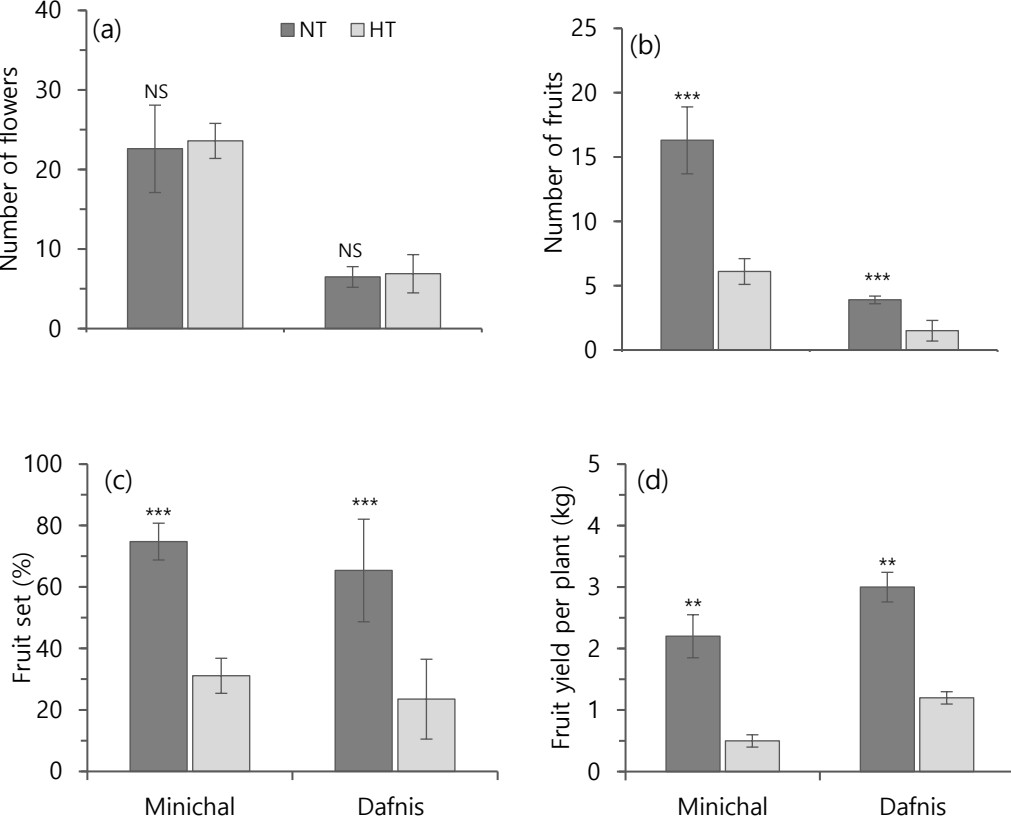

**Figure 9.** The number of flowers (**a**) and fruits (**b**), fruit set, (**c**) and fruit yield (**d**) of tomato cultivars grown in normal (NT) and high temperature (HT) greenhouses. Vertical bars represent ± SD (n = 6). NS, ** and *** indicate not significant and significant at the $p \leq 0.01$ and $p \leq 0.001$ levels in *t*-test, respectively.

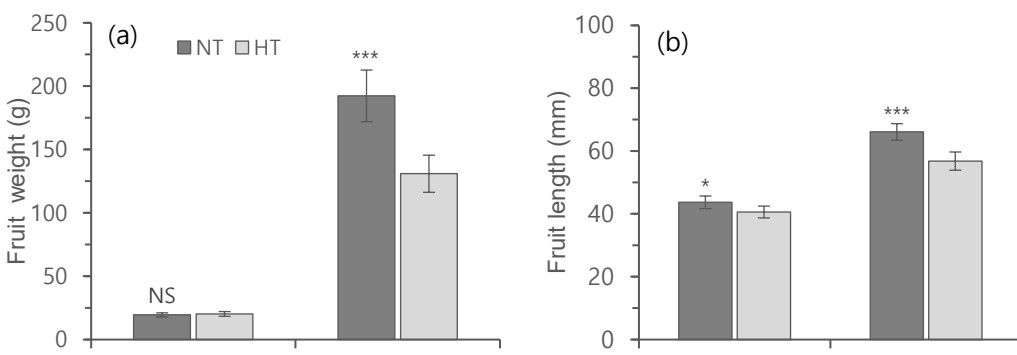

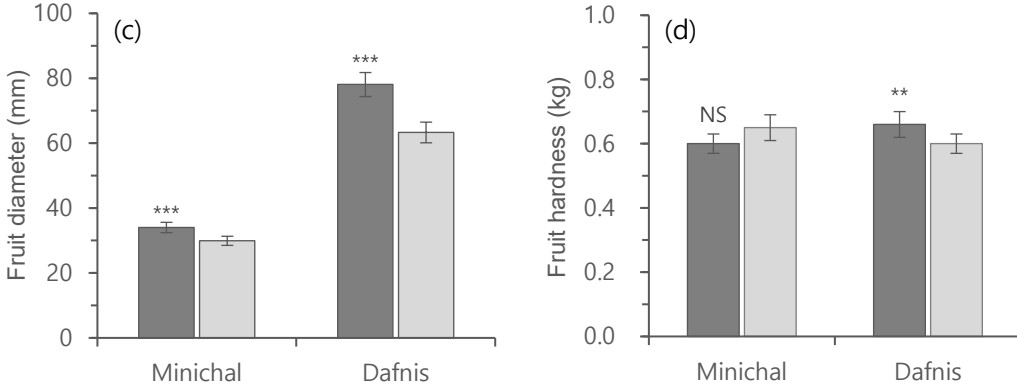

**Figure 10.** The fruit weight (**a**), fruit length (**b**), fruit diameter, (**c**) and fruit hardness (**d**) of tomatoes in normal (NT) and high temperature (HT) greenhouses. Vertical bars represent ± SD (n = 10). NS, *, ** and *** indicate not significant and significant at the $p \leq 0.05$, $p \leq 0.01$, and $p \leq 0.001$ levels in *t*-test, respectively.

The research showed the heat stress influence on the physiological, biochemical, vegetative, and reproductive parameters of tomato plants, but varied among the cultivars with contrasting heat tolerance level.

## 4. Discussion

Abiotic stress, such as high temperature stresses in tomato plants, is quite complex and demands multiple genotype evaluations to understand how the physiological parameters [4,12] and tolerance are altered at different growth stages [14,15,23]. The genotype behavior in growth stages of the plant is used as an extremely favorable method to estimate heat tolerance for the identification of tomato genotypes such as threshold temperature over 30 °C [3,24].

According to the present study, significant differences in heat tolerance among the seedlings of tomato cultivars appeared on stress day 3, where the symptoms of LHD levels increased more significantly in D than in M. Survival and the threshold level of high-temperature tolerance in the seedlings of D were identified on day 7, where the LHD level was over 60%, and it was screened as heat-susceptible, while M maintained stable heat tolerance with green leaves.

It is well known that high temperature adversely affects the physiological parameters of tomato plants, and consequently plant biomass production [10,16,25,26]. The chlorophyll index in tomato leaves decreased when the HT days increased, and the decrement was faster in heat-susceptible cultivar D than in tolerant cultivar M. Similar findings were reported in tomato cultivars [4,10,11], and the high content of CHL in heat-tolerant tomatoes gives better photosynthetic stability than heat-susceptible cultivars [2,10–12].

Regarding the differences among tomato cultivars, significant reduction of CHL in leaves has been identified in cultivar D, and it might be due to the decrease of chlorophyll *a* and *b* and carotenoid content, premature chlorosis, and withered leaves of tomato during heat stress condition [2,20]. On the other hand, leaves of tomato M were stay-green and magnitude change in the CHL was smaller than non-tolerant cultivars which was accordance with previous reports [10,27,28]. As mentioned above, the reduced $P_N$ value could be partially explained by the decreased leaf CHL content under heat stress as pigment content and composition are closely related to photosynthesis [29]. Green leaves in the heat-tolerant plants may contribute to the maintenance of yield at high temperatures [10,30], whereas decreased in $P_N$ may linked to the CHL reduction from the effect of reduced the pigments, altered chloroplast structure [9,31], carbohydrate synthesis [4,18] and low light condition during the heat stress [3, 29].

The present results showed that heat stress contributes to reduction of $P_N$ in tomato seedlings [18,20], but significant differences between tolerant and susceptible cultivars during the treatment period may persist [2,18,32]. Greener leaves with high CHL, $P_N$, $Gs$ and $Tr$ values were shown in the heat-tolerant tomato M when compared with heat-susceptible D during high-temperature conditions which may allow better leaf cooling and maintaining relatively high photosynthetic rate [4], whereas the heat-susceptible cultivars showed lowered contents with high leaf temperature [4,10,18].

Electrical conductivity test can reflect the stability of the cell membrane against abiotic stress in plants, where sub-optimal temperatures may alter the membrane structure of a plant cell, which leads to increased membrane permeability and certain small molecules within the cell flow out, causing an increase in electrical conductivity [17]. Therefore, it is a universally granted technique for measuring cell integrity with regarding cell membrane thermostability under stress condition [29,32–34].

A significant difference in values of EC under HT rather than NT conditions was found in both cultivars, but a significant increase in electrical leakage was determined in stressed tomato plants of D rather than heat-tolerant M. Several reports suggest that the high rate of EC in susceptible tomato plants indicates alterations in the permeability of the membranes and a reduction in their ability to retain solutes and water due to high temperature [3,35]. Moreover, in the tolerant tomato cultivar the alteration in the permeability of the membrane was not observed by heat-shock treatment, which indicates the maintenance of its functioning. However, we could not find the link between cell membrane thermo-stability to photosynthetic and transpiration rates [17].

Proline plays positive roles in enhancing plant tolerance to abiotic stress [36,37]. It stabilizes and protects the structure of enzymes and proteins, maintains membrane integrity, and scavenges reactive oxygen species [38]. Accumulation of PRL is considered a strong indicator of abiotic stress [38,39], but the high accumulation of PRL in plants during HT is detrimental for plant growth at recovery, which is not always a compatible solute during environmental stresses and high doses will impart toxic effects [25,36]. However, it may not always be used as a main indicator to measure the level of stress tolerance [38,40].

The finding of PRL content in heat-susceptible seedlings of D was significantly higher than in heat-tolerant M, which was positively linked to the EC rate during HT of seedlings [25]. Accumulation of PRL in tomato leaves increased when HT days increased in cultivar D, whereas M sowed steady values. Interestingly, this trend did not persist in the recovery period on day 8, where PRL showed an opposite trend, it reduced significantly in heat-susceptible tomato leaves of D than tolerant M. This indicates that heat-tolerant cultivar responded more to the duration of HT than the magnitude of leaf heat damage in terms of PRL accumulation, and according to this study we assume that low EC and PRL with stable rates contributed on keeping high rates of the photosynthetic rate at HT period and protect the impart toxic effects.

Regardless of tomato features, HT decreased the shoot fresh weight of seedlings compared to normal treatment [12], but PH and RFW rates significantly decreased in the heat-tolerant cultivar M compared to the heat-susceptible D. The present results could not provide links between vegetative parameters with values of $P_N$ and CHL during HT, where photosynthesis and CHL deficit may disrupt the metabolic pathways, reduce the growth rate and biomass whereas the tolerant genotypes accumulated more biomass, with lower heat injury index and higher fruit yield [4,10,12].

However, in contrast, different LHD levels of tomato plants significantly affect the vegetative parameters at recovery, where faster recovery was identified in a heat-tolerant cultivar M than susceptible one D. So, this is possibly due to the high CHL, $P_N$ and $Gs$ in a heat-tolerant cultivar throughout the entire days of HT and NT and dramatically high accumulation of the PRL at recovery than heat-susceptible cultivar D.

Comparison of the heat tolerance of tomato cultivars in the seedling stage with reproductive parameters such as NFL, NFR, FS, FY, FW, FL, FD, and FH under NT and HT

conditions showed that responses of reproductive traits to high temperature were different. There was no identified significant difference in flower production per truss among tomatoes in both conditions but some results have been reported [14], where NFL from control condition was positively correlated with NFL from HT, and NFL recommended as indicators of reproductive heat tolerance. FS and FY are important traits associated with heat tolerance and mainly used for tomato screening [16,33].

Analysis of the effect of HT on NFR and FS showed a significant reduction in both cultivars over NT plants. The results confirmed earlier findings that the reproductive growth stage is more sensitive to HT than the vegetative stage, where the adverse effects of high temperatures on FS primarily impact the viability of male and female gametes [14,16,25,33]. Although non-heat stressed pollen is not related to heat tolerance, as ascertained by percent FS, since high and low percent pollen germination can be identified in heat-susceptible and heat-tolerant tomato [13].

There were significant differences in FW and FH in cultivar D than in M but HT reduced FY, FL and FD in both tomato cultivars [16,25]. Tomato cultivars with small fruits were superior under HT conditions but the genotype may high FS and more flowers which is less affected by the heat stress than large size genotype [16,41,42], however they produced lower FY due to its smaller fruit size [25]. By contrast, the tomato genotypes accumulating higher PRL under HT likewise produced the highest FY [25], while in our study M had high accumulation PRL at recovery but produced lower FY than D, which showed high PRL in HT of seedlings.

The reproductive traits of tomato cultivars changes by the high temperature, since the mechanisms controlling heat stress response in plants are complex and response of genotypes as well as their physiological parameters on the high temperature is different and may vary in growth stages [14,18,25,38].

In general, we agreed with previous conclusion that the identification of heat-tolerant genotypes in a tomato breeding program should be deeply evaluated by growth stages, by fruit sizes and market preferences and a combination of heat-tolerant traits [3,15,16,18]. A heat-tolerant small-size tomato cultivar (such as M) in the seedling stage may produce less yield by heat stress rather than large size (on the example of D) in the reproductive stage, which was screened as heat-susceptible in seedling growth stage.

## 5. Conclusions

The present work showed that heat stress not only damages the appearance of tomato plants but also significantly affects the physiological-chemical and vegetative parameters in the seedlings stage. In the heat-susceptible tomato cultivar D were identified the reduction of the fruit parameters of plants such as fruit weight, fruit length and diameter, and fruit hardness by the high temperature; while in heat-tolerant cultivar M, the fruit weight and hardness increased but the fruit length and diameter decreased. Among small and large size tomatoes, there was no linkage between the seedling stage and growth stage in terms of heat tolerances with reproductive traits. To find out real tolerance, mechanism expression analysis of various gene products from tolerant genotypes selected according to screening at the vegetative and reproductive stages is necessary.

**Author Contributions:** Supervision, M.C.C.; methodology, E.Y.Y., S.Y.C.; formal analysis and investigation, S.N.R., E.Y.Y., H.B.J.; data curation, M.C.C., E.Y.Y.; writing—original draft preparation, S.N.R.; writing—review and editing, all authors. All authors have read and agreed to the published version of the manuscript.

**Funding:** This study was supported by a grant (Project No: PJ012662 "Breeding and selection of tomato lines with tolerance to abnormal temperatures") from the National Institute of Horticultural and Herbal Science, Rural Development Administration.

**Institutional Review Board Statement:** Not applicable.

**Informed Consent Statement:** Not applicable.

**Data Availability Statement:** The datasets generated during and/or analyzed during the current study are available from the corresponding author on responsible request.

**Conflicts of Interest:** The authors declare no conflict of interest.

## Abbreviations

HT—heat treatment, NT—normal treatment, D/N—day/night, LS—true leaf stage, LHD—leaf heat damage, $P_N$—the photosynthetic rate, $Gs$—stomatal conductance, $Ci$—intercellular $CO_2$ concentration, $Tr$—transpiration rate; PH—plant height; SFW—shoots fresh weight, RFW—roots fresh weight, NFL—number of flowers per truss, NFR—number of fruits per truss, FS—fruit set, FY—fruit yield, FW—fruit weight, FL—fruit length, FD—fruit diameter, FH—fruit hardness.

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
