# Peer review of "Heat Treatment in Two Tomato Cultivars: A Study of the Effect on Physiological and Growth Recovery"

_horticulturae, doi:10.3390/horticulturae7050119_

Round 1

Reviewer 1 Report

The paper in its current form cannot be considered for publication. First of all, the statistical design must be clarified: in particular, were the same plants or plants chosen at random used for sampling? Only by clarifying the methodological aspects could the very limited number of plants analyzed be accepted. There are also some inconsistencies in the text: for example, expressing the doses of fertilizer used in terms of kg ha-1 is not appropriate given the limited number of plants under examination (it is more logical to report this information in g m-2). The extensive bibliography reported must be carefully double checked because in some cases there are inconsistencies; for example the citations 34, 62 and 61 are reported in the text in a position other than the progressive number and the citation 57 reported in the bibliography does not appear in the text. 
Moving on to consider more general problems, in lines 385-388 the Authors affirm, citing reference 22 as confirmation: "Moreover tomato breeding programs for high temperature tolerance to be consider different selection criteria depending on fruit types with different sizes of the target cultivar "; apart from the English form that should certainly be improved, on the basis of what is reported in the text it is not clear why cultivars tolerant and susceptible to high temperatures but characterized by different types of fruit have been taken into consideration. More generally, the limited number of plants used, belonging to only two cvs, for a species of great economic importance such as Solanum lycopersicum, provides data of some interest but certainly not fundamental for tomato breeding. 
Furthermore, the way of developing the Discussion is questionable in which it appears that much of what is reported in the paper is nothing more than a confirmation of what has already been indicated by other Authors, at least based on the bibliographical references reported at the end of individual sentences. If this is true, the paper could be fine for a Master's thesis but not for a scientific publication, otherwise the discussion and bibliographic citations would have to be suitably reset. 
Finally, should the paper be re-proposed for a possible publication, the English text which is not easy to understand in the present form must be carefully revised.

Author Response

Response sheet
Reviewer 1
We appreciate the time and effort of the reviewer 1. Comments and suggestions helped to 
improve present manuscript and compared to the original. We accommodated all of your
comments, thoroughly rewrote and reorganized the manuscript. We hope the manuscript after 
the revising is fit for the publication in “Horticulturae” journal’s special issues “Factors 
affecting the quality, and shelf life of horticultural crops”. 
Authors’ responses
 The statistical design must be clarified: in particular, were the same plants or plants 
chosen at random used for sampling?
☞ We agree with reviewer’s comments that some missing in Material and Methods part. We 
have reorganized and rewritten materials and methods part of MS. (Line- 70-170)
 There are also some inconsistencies in the text: for example, expressing the doses of 
fertilizer used in terms of kg ha-1 is not appropriate given the limited number of plants
under examination (it is more logical to report this information in g m2
)
☞ We agreed with your comments and we have rewritten in the text (Line 150-153).
 The extensive bibliography reported must be carefully double checked because in 
some cases there are inconsistencies; for example the citations 34, 62 and 61 are 
reported in the text in a position other than the progressive number and the citation 57 
reported in the bibliography does not appear in the text. 
☞ Thank you for finding, citations and the bibliography have rewritten and reorganized in the 
text.
 In lines 385-388 the Authors affirm, citing reference 22 as confirmation: "Moreover 
tomato breeding programs for high temperature tolerance to be consider different 
selection criteria depending on fruit types with different sizes of the target cultivar "; 
apart from the English form that should certainly be improved, on the basis of what is 
reported in the text it is not clear why cultivars tolerant and susceptible to high temperatures but characterized by different types of fruit have been taken into 
consideration. 
☞ We have rewritten and reorganized your comments in the text. (Line 403-408). 
 The limited number of plants used, belonging to only two cvs, for a species of great 
economic importance such as Solanum lycopersicum, provides data of some interest 
but certainly not fundamental for tomato breeding. 
☞ Our purpose as mentioned above was to find out survival rate and physiological responses 
of seedlings of tomatoes, one tolerant and the other susceptible to high temperature (severe day 
and night temperature regime 40°C), and effect of leaf heat damage rates (LHD) on subsequent 
vegetative traits at recovery and identifies the connection of heat tolerance rate in seedling 
stage with reproductive traits at growth stage.
 Finally, should the paper be re-proposed for a possible publication, the English text 
which is not easy to understand in the present form must be carefully revised.
☞ We agree with your comments and manuscript have reorganized and revised and edited in 
English

Reviewer 2 Report

Title

I recommend you review your title to have more impact. An idea could be: " Heat treatment in two tomatoes varieties: study of effect on a physiological and growth recovery"

Abstact

Line 12 please change "… seriously effects…" with "… significantly affects"

This section requires a complete review and edit by someone who has better English and grammar skills.

Introduction

This section requires a complete review and edit by someone who has better English and grammar skills. I suggest reviewing, the organization and structure of the introduction, it is not exhaustive.

Why the choose of "Dafnis" and "Minichal" varieties?

Author Response

Response sheet
Reviewer 2
One more we appreciate the time and effort of the reviewer 2. Comments and suggestions 
helped to improve present manuscript and compared to the original, thoroughly rewrote and 
reorganized the manuscript and it is in much better shape than the original. Below we have 
provided the answers for your comments. We hope the manuscript after the revising is fit for 
the publication in n special issues of “Horticulturae” journal “Factors affecting the quality, and 
shelf life of horticultural crops”. 
Authors’ responses
 I recommend you review your title to have more impact. An idea could be: " Heat 
treatment in two tomatoes varieties: study of effect on a physiological and growth 
recovery"
.☞ We agree fully with your comment and we have rewritten the title according to your 
suggestion. 
 Line 12 please change "… seriously effects…" with "… significantly affects"
.☞ Thank you for recommendation, it was corrected in abstract (Line 11). 
 Abstract section requires a complete review and edit by someone who has better 
English and grammar skills.
.☞ We agree with your comment and abstract was rewritten and edited in English. (Line 11-
26)
 Introduction section requires a section requires a complete review and edit by someone 
who has better English and grammar skills. I suggest reviewing, the organization and 
structure of the introduction, it is not exhaustive.
.☞ We agree with your comment. Introduction section was rewritten and edited in English 
(Line 30-67).
 Why the choose of "Dafnis" and "Minichal" varieties?
.☞Both cultivars is commercial and widely cultivating in South Korea, and identified with 
different heat tolerance in the seedling stage.

Reviewer 3 Report

The manuscript "Status of two tomatoes varieties with different heat susceptibility during heat treatment on physiological and growth recovery" is proposing to study the influence of high temperatures on two different tomato cultivars, “Dafnis” and “Minichal”.

First of all, the paper is already published as "preprint" in two places, but according to Journal's policy, "Horticulturae accepts submissions that have previously been made available as preprints provided that they have not undergone peer review. A preprint is a draft version of a paper made available online before submission to a journal." Links below:

https://www.preprints.org/manuscript/202102.0122/v1

https://www.researchgate.net/publication/349052420_Effect_of_heat_treatment_on_physiological_and_recovery_growth_status_of_two_tomato_cultivars_with_different_heat_susceptibility

The paper requires major revision in order to clarify the following aspects:

Abstract

The Abstract should be more concise, of maximum 200 words, and should follow the Journal's recommended style.

Performing the study at only one temperature, 40°C, is not sufficient to assess "the interrelation heat tolerance screening methods between seedling and reproductive stages of tomatoes".

Phrases are intricate and should be shorten and clarified. For example, in Lines 22-32 it is not clear the effect of heat on both tomato cultivars.

Introduction

The Introduction is short and too general. Aspects regarding abiotic stress of different tomato cultivars could be discussed in terms of analyses, plant protection and remediation.

Materials and methods

2.1. After seedlings transfer in growth chamber at 40°C, it is not mentioned the control (at normal temperature). Also, studying the effect of only one high temperature is not so relevant. At least 2 intermediary temperatures between 25-40°C should be tested.

Light intensity of 800μmol m-²s-¹ is considered by some studies to be a stress value. This fact should be discussed and other intensities could be also tested. "Heat tolerance treatment (HT)" cannot be "continued seven days", instead probably the heat treatment. Water quantity should be also provided.

It is not clear the fertilization used: "… manure at a dose of 10,000 kg ha-1 and basal fertilizer containing 160 kg ha-1 N, 80 kg ha-1 K2O, 160 kg ha-1 P2O5 and regularly watered to avoid drought and fertilized on a weekly basis." 10 t ha-1 manure? What elemental composition? Additional NPK dosages seem very high compared to recommended dosages by FAO (http://www.fao.org/land-water/databases-and-software/crop-information/tomato/en/). Please discuss.

2.3.  Please explain why the chlorophyll index  was measured on day 0,1,3,5 and 7, while photosynthesis parameters only on days 1 and 3. What irradiation levels were used for photosynthetic rate measurement?

2.7. It is not clear what were the normal growth conditions and the number of days - Lines144-145: "All heat treated seedlings from grades 3, 4, and 5 and from control (no treated) were transplanted into pots after 3 days of recovery in greenhouse NT condition (D/N 30-32/22-24°C) for 28 days."

Results

Lines 193-194 are ambiguous. Please clarify.

Fig. 4, 5,6. The values for day 5 and 7 could be also of interest.

In Line 85 it is mentioned that 6 plants were prepared for reproducibility. Then, why n=4 in Fig 1 and n=3 in Fig 3,4,5,6?

Measurements after plants' recovery in green house in NT conditions could be also relevant.

Discussion

The novelty of the study should be emphasized.

Phrase L288-292 is too long and ambiguous.

L322: I think is "reflex" instead of "reflects".

L341-342: PRL content is higher or similar?

Author Response

Response sheet
Reviewer 3
We appreciate the time and effort of the reviewer 3. Comments and suggestions really helped 
to improve present manuscript and compared to the original. We accommodated all of your
comments, thoroughly rewrote and reorganized the manuscript. We hope the manuscript after 
the revising is fit for the publication in special issues of “Horticulturae” journal “Factors 
affecting the quality, and shelf life of horticultural crops”. 
Authors’ responses
 The Abstract should be more concise, of maximum 200 words, and should follow the 
Journal's recommended style.
☞ We agree with your comments and we have rewritten of the abstract section (Line 11-26)
 Performing the study at only one temperature, 40°C, is not sufficient to assess "the 
interrelation heat tolerance screening methods between seedling and reproductive 
stages of tomatoes".
☞ Well known that identifying the mechanism of heat tolerance among plants is comlex
manner. Therefore, the purpose of this research is to analyze the survival rates and 
physiological responses in the seedlings and adult plants of tomato cultivars, with contrasting 
heat tolerance level. In the present study tolerant and susceptible tomato cultivars were treated 
with severe day and night temperature regime of 40°C, and effect of leaf heat damage rates 
(LHD) on subsequent vegetative traits at recovery and identifies the connection of heat 
tolerance rate in seedling stage with reproductive traits at the growth stage were investigated.
 Phrases are intricate and should be shorten and clarified. For example, in Lines 22-32 
it is not clear the effect of heat on both tomato cultivars.
☞ We agree with your comments and we have rewritten and reorganized of the abstract 
section (Line 11-26)
 The Introduction is short and too general. Aspects regarding abiotic stress of different 
tomato cultivars could be discussed in terms of analyses, plant protection and 
remediation.
☞ We have rewritten of the Introduction section, but we would like to inform that in this 
section we tried to use just common sentences, since in Discussion part we quite described the 
our results in line with other references. (Line 30-67). 2.1. After seedlings transfer in growth chamber at 40°C, it is not mentioned the control 
(at normal temperature). Also, studying the effect of only one high temperature is not 
so relevant. At least 2 intermediary temperatures between 25-40°C should be tested.
☞ As mentioned above we used severe heat stress regime and tried to understand the 
physiological responses of two tomatoes with different heat tolerance rate. 
 Light intensity of 800μmol m-²s-¹ is considered by some studies to be a stress value. 
This fact should be discussed and other intensities could be also tested. "Heat tolerance 
treatment (HT)" cannot be "continued seven days", instead probably the heat treatment. 
Water quantity should be also provided.
☞ According to literatures responses of crops on light intensity is different depends on growth 
stages and as mentioned above we used severe heat stress regime. We agree with you and we 
corrected the text “Heat tolerance treatment (HT) into the heat treatment. Also, we provided 
water quantity in Materials and methods section (Line 75-83)
 It is not clear the fertilization used: "… manure at a dose of 10,000 kg ha-1 and basal 
fertilizer containing 160 kg ha-1 N, 80 kg ha-1 K2O, 160 kg ha-1 P2O5 and regularly 
watered to avoid drought and fertilized on a weekly basis." 10 t ha-1 manure? What 
elemental composition? Additional NPK dosages seem very high compared to 
recommended dosages by FAO (http://www.fao.org/land-water/databases-andsoftware/crop-information/tomato/en/).
☞ We have to explain that depends on the climatic and soil conditions the fertilization rate 
can be varied around the world. In our study, the soil in two greenhouses were prepared 
according to the recommendations of Korea soil information system [https://soil.rda.go.kr
Korea soil information system] equally with pre-plant broadcast manure at a dose of 1 kg m2
(Bio Sangto, Seoul, Korea; containing coco peat- 47.2%, peat moss- 35%, zeolite- 7%, 
vermiculite- 10.0%, dolomite- 0.6%, humectant-0.006% and fertilizers- 0.194%) and basal 
fertilizer containing 16 g m2 N, 8 g m2 K2O, 16 g m-2 P2O5 (Daeyu, Mulpure) and regularly 
watered to avoid drought and fertilized weekly basis N-6, P-10 and K-5 (HYPONeXm, Japan)
(Line 73-75, 150-153).
 2.3. Please explain why the chlorophyll index was measured on day 0,1,3,5 and 7, 
while photosynthesis parameters only on days 1 and 3. What irradiation levels were 
used for photosynthetic rate measurement?
☞ Photosynthesis parameters were measured on days 0, 1 and 3. Unfortunately we had an 
only one the photosynthesis machine and it was occupied that days of HT 5 and 7, therefore 
we presented data only before start to heat treatment day 0 (initial rate) and on days 1 and 3 at stress regime. But, according to literatures actually short period of heat treatment is enough to 
know response of plants. Whereas, chlorophyll was estimated by SPAD meter during 0, 1, 3, 
5 and 7 days of heat treatment. Light response curves (PAR) was set to 800 μmol m2
s1
 2.7. It is not clear what were the normal growth conditions and the number of days -
Lines144-145: "All heat treated seedlings from grades 3, 4, and 5 and from control (no 
treated) were transplanted into pots after 3 days of recovery in greenhouse NT 
condition (D/N 30-32/22-24°C) for 28 days."
☞ We agree with your comments and we have rewritten and reorganized in the text. (Line 
130-137).
 Lines 193-194 are ambiguous. Please clarify.
☞ We agree with your comments and we have rewritten in the text. (Line 198-201).
 In Line 85 it is mentioned that 6 plants were prepared for reproducibility. Then, why 
n=4 in Fig 1 and n=3 in Fig 3,4,5,6? Measurements after plants' recovery in green 
house in NT conditions could be also relevant.
☞ We agree with your comments and there were misunderstanding in writing the materials 
and methods section. We have rewritten and reorganized the materials and methods section in 
the text. For information, for each cultivar in Fig 1, a 4 technical replications (a total 32 
seedlings) were heat treated in the growth chamber for 7 days (Line 75-78). In Fig. 3-6 were 
used three biological replications. And, in the next study “Experiment 2” we used 6 biological 
plants (replications for each cultivars) at NT and HT conditions, respectively.
 Phrase L288-292 is too long and ambiguous.
☞ We agree with your comment and we have rewritten and reorganized text in the discussion 
section (Line 294-298)
 L322: I think is "reflex" instead of "reflects".
☞ We agree with your comment and we have rewritten in the text of discussion section (Line 
330) L341-342: PRL content is higher or similar?.
☞ We agree with your comment, where it was written unclear and we have rewritten in the 
text of discussion section (Line 348-352

Round 2

Reviewer 3 Report

The Authors improved the initial version of the manuscript. Though, the English still needs to be revised. Some examples in version 2:

Line 38: The predicate of the sentence is missing. 

L49: "multiple" instead of "mutiple"

L51-52: It is not clear what the results exhibited.

L55: "fruits" instead of "frutis"

L63-67: Please rephrase.

L87: "by" instead of "b"

L235: "different" instead of "differed"

L288-289, 294-298: Please revise.

L305: "It is" well known that...

L330: "test can reflect..."

L346: "It stabilizes and protects..."

L361: "... contributed on keeping high rates..."

L391-395, 400-402: Please rephrase.

Author Response

Please see attached file below, thank you
